# Myogenic vasoconstriction requires $G_{12}/G_{13}$ and LARG to maintain local and systemic vascular resistance

Ramesh Chennupati[1], Angela Wirth[2], Julie Favre[3], Rui Li[1], Rémy Bonnavion[1], Young-June Jin[1], Astrid Wietelmann[4], Frank Schweda[5], Nina Wettschureck[1,6,7], Daniel Henrion[3], Stefan Offermanns[1,6,7]*

[1]Department of Pharmacology, Max Planck Institute for Heart and Lung Research, Bad Nauheim, Germany; [2]Institute of Pharmacology, University of Heidelberg, Heidelberg, Germany; [3]Laboratoire MITOVASC, UMR CNRS 6015 - INSERM 1083, Université d'Angers, Angers, France; [4]Scientific Service Group Nuclear Magnetic Resonance Imaging, Max Planck Institute for Heart and Lung Research, Bad Nauheim, Germany; [5]Institute of Physiology, University of Regensburg, Regensburg, Germany; [6]Centre for Molecular Medicine, Medical Faculty, JW Goethe University Frankfurt, Frankfurt, Germany; [7]German Center for Cardiovascular Research (DZHK), Berlin, Germany

**Abstract** Myogenic vasoconstriction is an autoregulatory function of small arteries. Recently, G-protein-coupled receptors have been involved in myogenic vasoconstriction, but the downstream signalling mechanisms and the in-vivo-function of this myogenic autoregulation are poorly understood. Here, we show that small arteries from mice with smooth muscle-specific loss of $G_{12}/G_{13}$ or the Rho guanine nucleotide exchange factor ARHGEF12 have lost myogenic vasoconstriction. This defect was accompanied by loss of RhoA activation, while vessels showed normal increases in intracellular $[Ca^{2+}]$. In the absence of myogenic vasoconstriction, perfusion of peripheral organs was increased, systemic vascular resistance was reduced and cardiac output and left ventricular mass were increased. In addition, animals with defective myogenic vasoconstriction showed aggravated hypotension in response to endotoxin. We conclude that $G_{12}/G_{13}$- and Rho-mediated signaling plays a key role in myogenic vasoconstriction and that myogenic tone is required to maintain local and systemic vascular resistance under physiological and pathological condition.
DOI: https://doi.org/10.7554/eLife.49374.001

*For correspondence:
stefan.offermanns@mpi-bn.mpg.de

Competing interests: The authors declare that no competing interests exist.

## Introduction

The tone of arterial blood vessels controls blood flow and is constantly regulated by multiple stimuli including the blood flow itself as well as various neural and humoral stimuli. A particular form of autoregulation, the myogenic response, was first described by *Bayliss (1902)*, who observed that stretch imposed on the vascular wall by an increase in intraluminal pressure induced a contraction of the vessel. This ability of vascular smooth muscle cells to respond to mechanical stretch has been observed in a variety of vascular beds. It has been suggested that myogenic vasoconstriction is involved in maintaining basal vascular tone to promote constant perfusion despite fluctuations in perfusion pressure and to prevent tissue damage in cases of overperfusion (*Davis and Hill, 1999*; *Loutzenhiser et al., 2002*; *Walsh and Cole, 2013*; *Carlström et al., 2015*). However, in vivo evidence for this is still lacking.

As any vascular smooth muscle contraction, the myogenic response goes along with phosphorylation of the myosin light chain (MLC), which allows myosin to interact with actin and to generate contractile forces (*Somlyo et al., 2004*; *Vetterkind and Morgan, 2012*). MLC phosphorylation can be increased by activation of the MLC kinase (MLCK) through a $Ca^{2+}$- and calmodulin-dependent mechanism or by inhibition of myosin phosphatase via the Rho/Rho-kinase pathway (*Somlyo and Somlyo, 2000*). How these two pathways are regulated during myogenic constriction of vascular smooth muscle cells is incompletely understood. Both stretch-activated ion channels and mechanically activated G-protein-coupled receptors (GPCRs) have been shown to play a role in the initiation of the myogenic response (*Kauffenstein et al., 2012*; *Schubert et al., 2008*; *Mederos Y Schnitzler et al., 2016*). It is widely believed that induction of vascular myogenic contraction requires membrane depolarization through activation of mechanically activated ion channels which would then induce opening of voltage-dependent calcium channels resulting in an increase in intracellular calcium concentration (*Davis and Hill, 1999*; *Kauffenstein et al., 2012*; *Knot and Nelson, 1998*; *Welsh et al., 2002*; *Jensen et al., 2017*; *Hill et al., 2010*). Although several ion channels have been involved in stretch-induced depolarization, including the transient receptor potential (TRP) channels TRPC6 and TRPM4, the molecular nature of the channel responsible for stretch-induced depolarization of vascular smooth muscle cells has remained elusive (*Kauffenstein et al., 2012*; *Earley and Brayden, 2015*).

More recently, several GPCRs have been involved in the myogenic response of vascular smooth muscle cells (*Kauffenstein et al., 2012*; *Mederos Y Schnitzler et al., 2016*). These include the purinergic $P2Y_6$ receptor, the thromboxane $A_2$ (TP) receptor, sphingosine-1-phosphate receptors as well as the angiotensin $AT_1$ receptor and the cysteinyl leukotriene one receptors ($CysLT_1Rs$) (*Mederos y Schnitzler et al., 2008*; *Schleifenbaum et al., 2014*; *Storch et al., 2015*; *Kauffenstein et al., 2016*; *Kroetsch and Bolz, 2013*). While stretch-induced activation of $P2Y_6$, TP and S1P receptors is believed to involve the release of agonistic ligands (*Kauffenstein et al., 2012*; *Kauffenstein et al., 2016*; *Kroetsch and Bolz, 2013*), evidence has been provided that mechanical activation of $AT_1$ and $CysLT_1$ receptors occurs in an agonist-independent manner (*Mederos Y Schnitzler et al., 2016*; *Mederos y Schnitzler et al., 2008*; *Storch et al., 2015*). The GPCRs implicated in vascular myogenic contraction can couple to several G-proteins including $G_q/G_{11}$ and $G_{12}/G_{13}$ (*Kauffenstein et al., 2012*). Previous work has shown that in vascular smooth muscle cells coupling of receptors to $G_q/G_{11}$ through stimulation of phospholipase C-β results in the $Ca^{2+}$- and calmodulin-dependent activation of MLCK, whereas coupling to $G_{12}/G_{13}$ via activation of the Rho/Rho-kinase pathway results in the inhibition of myosin phosphatase and that these two pathways can synergize (*Gohla et al., 2000*; *Somlyo and Somlyo, 2003*; *Wirth et al., 2008*). Myogenic contraction can be strongly reduced by pharmacological inhibition of Rho-kinase downstream of $G_{12}/G_{13}$ (*Schubert et al., 2008*; *Li and Brayden, 2017*; *Moreno-Domínguez et al., 2013*; *Schubert et al., 2002*; *Gokina et al., 2005*) as well as of $G_q/G_{11}$ (*Storch et al., 2015*). However, since these approaches also affect the function of other vascular cells, including endothelial cells, and since pharmacological tools are in many cases not specific (*Bain et al., 2007*; *Gao and Jacobson, 2016*), the role of the two major G-protein-mediated signaling transduction pathways inducing vascular smooth muscle contraction in myogenic tone regulation remains unclear.

To understand the function of $G_q/G_{11}$- and $G_{12}/G_{13}$-mediated signaling pathways in vascular myogenic contraction and to better understand the role of myogenic vasoconstriction in vivo, we studied vessels and animals with inducible smooth muscle-specific $G\alpha_q/G\alpha_{11}$ and $G\alpha_{12}/G\alpha_{13}$ deficiency. Using in vitro and in vivo experiments, we found that $G_{12}/G_{13}$ and the Rho guanine nucleotide exchange factor ARHGEF12 are required for the myogenic response. By using mice lacking $G_{12}/G_{13}$ or ARHGEF12 in smooth muscle cells as a model for defective myogenic vasoconstriction, we found that myogenic tone is critical for maintaining basal peripheral resistance and proper organ perfusion under physiological and pathophysiological conditions.

## Results

### Myogenic autoregulation in arteries depends on smooth muscle $G_{12}/G_{13}$ function

To analyze the role of $G_q/G_{11}$ and $G_{12}/G_{13}$ in vascular myogenic contraction, we used tamoxifen-inducible smooth muscle-specific $G\alpha_q/G\alpha_{11}$- and $G\alpha_{12}/G\alpha_{13}$-deficient mice described before (*Wirth et al., 2008*), henceforth called Sm-q/11-KO and Sm-12/13-KO, respectively. To analyze myogenic contraction in these animals, we performed pressure myography on first or second order mesenteric arteries from wild-type, Sm-q/11-KO and Sm-12/13-KO mice treated for five days with tamoxifen. Mesenteric arteries were exposed to stepwise (20 mmHg) increases in intraluminal pressure between 20 and 140 mmHg in the presence and absence of extracellular $Ca^{2+}$ to determine active and passive vessel diameter, respectively. A difference between the diameter in the presence and absence of extracellular $Ca^{2+}$ at each pressure step was defined as an active vasoconstriction. Stepwise increases in intraluminal pressure generated active contraction at pressures higher than 60 mmHg in mesenteric arteries from wild-type mice reaching peak contractions at about 80 to 120 mmHg (*Figure 1A and B* and Suppl. *Figure 1A*). Mesenteric arteries from Sm-q/11-KO mice behaved like wild-type mice and showed comparable myogenic contraction (*Figure 1A* and *Figure 1—figure supplement 1A and B*). In contrast, mesenteric arteries from Sm-12/13-KO mice completely lacked active vasoconstriction indicating loss of myogenic contraction (*Figure 1B* and *Figure 1—figure supplement 1C*), a finding also made in third order mesenteric arteries (*Figure 1—figure supplement 1E–G*).

We also performed myogenic tone experiments in cerebral vessels (*Figure 1C and D*). When proximal cerebral arteries were exposed to stepwise increases in intraluminal pressure, an active vasoconstriction was observed in vessels of wild-type mice, which was indistinguishable from the response observed in vessels from mice with induced smooth muscle-specific $G\alpha_q/G\alpha_{11}$ deficiency (*Figure 1C*). However, the myogenic response in vessels from induced smooth muscle-specific $G\alpha_{12}/G\alpha_{13}$-deficient animals was significantly reduced (*Figure 1D*). Thus, similar to mesenteric vessels, the myogenic response of cerebral vessels does not require $G_q/G_{11}$-mediated signaling in smooth muscle cells but depends on a functional $G_{12}/G_{13}$-mediated signaling pathway.

### Smooth muscle ARHGEF12 and RhoA-mediated signaling are involved in myogenic contraction

Given that $G_{12}/G_{13}$ can couple GPCRs to the regulation of Rho/Rho-kinase (*Worzfeld et al., 2008*; *Kozasa et al., 2011*), we determined RhoA activity during the development of myogenic tone in mesenteric vessels from wild-type, Sm-q/11-KO and Sm-12/13-KO mice. In pressurized mesenteric arteries from Sm-12/13-KO mice, RhoA activity was strongly reduced compared to vessels from wild-type and Sm-q/11-KO mice (*Figure 1E*). Consistent with published data (*Gokina et al., 2005*; *Dubroca et al., 2005*), we found that the Rho-kinase inhibitor Y-27632 blocked myogenic response in mesenteric arteries (*Figure 1F*). In addition, we found that phosphorylation of the regulatory subunit of myosin phosphatase, MYPT1, a substrate of Rho-kinase, was increased in pressurized wild-type mesenteric arteries, but in vessels from Sm-12/13-KO mice this effect could not be observed any more (*Figure 1G*). This supports the concept that Rho/Rho-kinase-mediated signaling plays a central role in the myogenic response. The Rho-guanine nucleotide exchange factor 12 (ARHGEF12, also known as leukemia-associated RhoGEF (LARG)) has been shown to couple $G_{12}/G_{13}$ to the regulation of RhoA activity in vascular smooth muscle cells (*Wirth et al., 2008*). We therefore tested myogenic contraction in mesenteric arteries from tamoxifen-induced smooth muscle-specific ARHGEF12-deficient animals (Sm-Larg-KO) and found that, similar to vessels from Sm-12/13-KO mice, RhoA activation was decreased (*Figure 1E*), and the myogenic response was abrogated (*Figure 1H* and *Figure 1—figure supplement 1D*). These data strongly indicate that $G_{12}/G_{13}$-mediated Rho/Rho-kinase activation is required for myogenic contraction.

To test whether the increase in intracellular $[Ca^{2+}]$ described to occur during myogenic contraction (*Gokina et al., 2005*; *D'Angelo et al., 1997*) was affected by smooth muscle-specific $G\alpha_q/G\alpha_{11}$ or $G\alpha_{12}/G\alpha_{13}$ deficiency, we loaded mesenteric arteries with Fura-2AM and determined the intracellular free $Ca^{2+}$ concentration during the stepwise increases in intraluminal pressure (*Figure 1I–K*). We observed in vessels from wild-type mice an increase in $[Ca^{2+}]_i$, which accompanied the myogenic

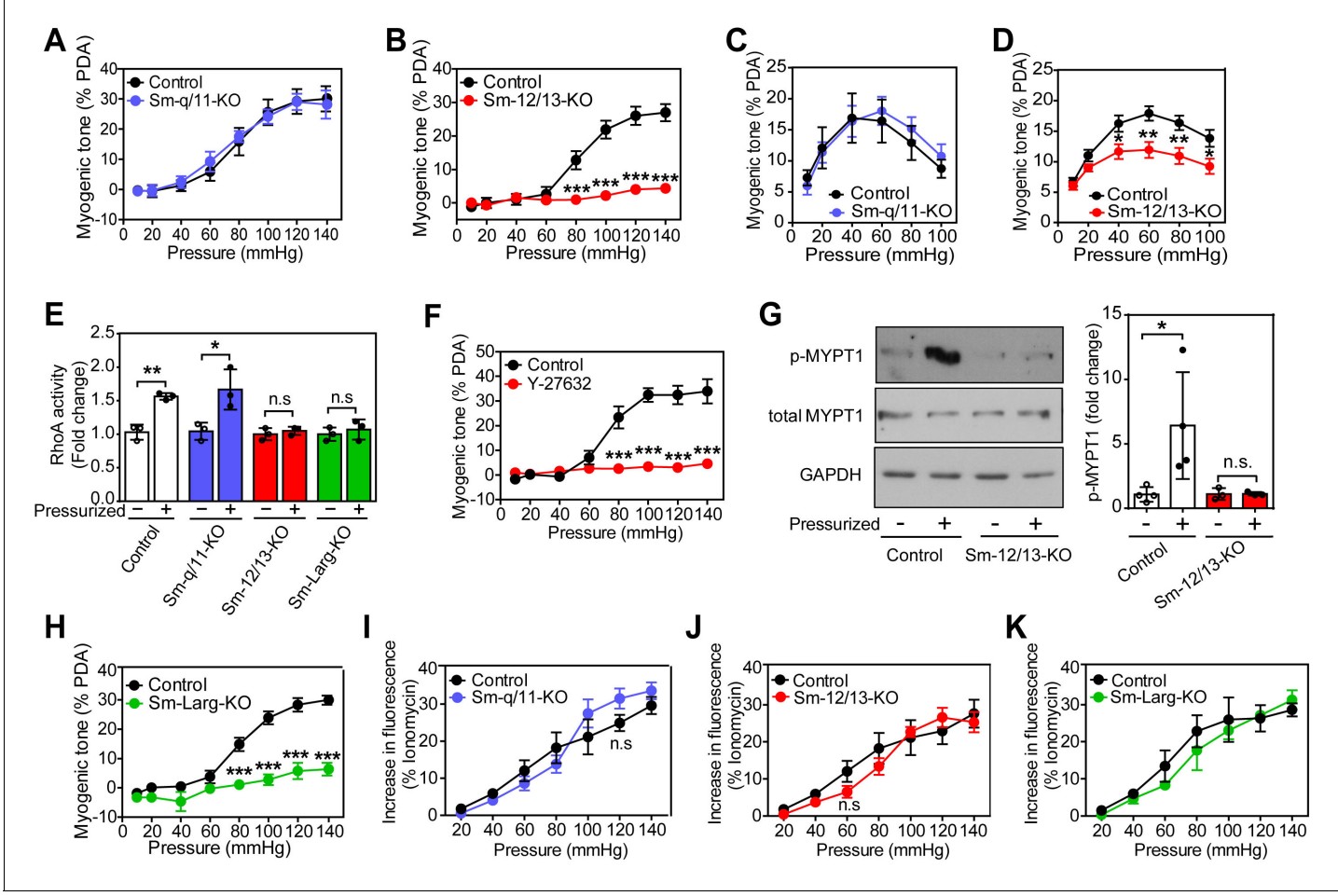

**Figure 1.** Role of $G_{12}/G_{13}$, ARHGEF12 (LARG) and RhoA signaling in myogenic contraction. (**A–D**) First order mesenteric arteries (**A,B**) and posterior cerebral arteries (**C,D**) were isolated from control (black circles), Sm-q/11-KO (blue circles) and Sm-12/13-KO (red circles) mice and were pressurized with the indicated pressure steps. The myogenic tone (% of passive diameter (PDA)) was determined using MyoView four software (**A,B**) or with Acknowledge software (**C,D**) (n = 7 mice (control groups and Sm-q/11-KO) and eight mice (Sm-12/13-KO) in A and B and n = 5 mice (control group for Sm-q/11-KO), nine mice (Sm-q/11-KO), 27 mice (control group for Sm-12/13-KO) and 21 mice (Sm-12/13-KO) in C and D). (**E**) RhoA activity was determined in the pressurized (80 mmHg for 10 min, '+') and non-pressurized ('-') mesenteric arterial segments (6–8 per animal) from control (open bars), Sm-q/11-KO (blue bars), Sm-12/13-KO (red bars) and Sm-Larg-KO (green bars) mice (n = 3 mice in each group). (**F**) Myogenic tone responses of first order mesenteric arteries incubated for 30 min in the presence (red circles) or absence (black circles) of Rho-kinase inhibitor (Y27632, 10 µM) (n = 3 mice for both groups). (**G**) Mouse MYPT1 phosphorylation (P-MYPT1) was determined in pressurized (80 mmHg '+') and non-pressurized ('-') mesenteric arteries from control and Sm-12/13-KO mice after homogenization by immunoblotting. Total MYPT1 and GAPDH immunoblots served as controls. The bar diagram shows a statistical analysis of immunoblots (n = 8 mice in four independent experiments (control) and n = 6 mice in three independent experiments (Sm-12/13-KO)). (**H**) Myogenic tone responses of first order mesenteric arteries in control (black circles) and Sm-Larg-KO (green circles) mice (n = 6 mice (control) and five mice (Sm-Larg-KO)). (**I–K**) Determination of intracellular [Ca²⁺] during myogenic tone induction in first order mesenteric arteries isolated from control (black circles), Sm-q/11-KO, (blue circles), Sm-12/13-KO (red circles), and Sm-Larg-KO mice (green circles). All arteries were loaded with Fura-2-AM (12.5 µM) for determination of free intracellular [Ca²⁺] (n = 5 mice (control groups, Sm-q/11-KO and Sm-12/13-KO) and n = 4 mice (Sm-Larg-KO)). All values are mean values ± SEM. *, $p \leq 0.05$; **, $p \leq 0.01$; ***, $p \leq 0.001$; n.s. = non significant (2-way ANOVA and Bonferroni's post-hoc test (in A-D and F-K); unpaired t-test (in E)).

DOI: https://doi.org/10.7554/eLife.49374.002

The following source data and figure supplements are available for figure 1:

**Source data 1.** Analysis of pressure-induced changes in myogenic tone, RhoA-activation, Rock inhibition and intracellular calcium release in smooth muscle-specific $G\alpha_q/G\alpha 1_{11}$, $G\alpha_{12}/G\alpha_{13}$, and ARHGEF12 (LARG) deficient mice.
DOI: https://doi.org/10.7554/eLife.49374.005
**Source data 2.** Western blot pictures (uncut) of pressure-inducedP-MYPT1 phosphorylation in smooth muscle-specificGα12/Gα13 deficient mice.
DOI: https://doi.org/10.7554/eLife.49374.006
**Figure supplement 1.** Effect of vascular smooth muscle-specific Gαq/Gα11, Gα12/Gα13, and ARHGEF12 (LARG) deficiency on myogenic tone responses.

*Figure 1 continued on next page*

*Figure 1 continued*

DOI: https://doi.org/10.7554/eLife.49374.003

**Figure supplement 2.** Effect of L- type and TRP channel inhibitors on myogenic tone.

DOI: https://doi.org/10.7554/eLife.49374.004

response. However, this response was unaffected by smooth muscle-specific $G\alpha_q/G\alpha_{11}$ and $G\alpha_{12}/G\alpha_{13}$ deficiency (*Figure 1I–K*), whereas inhibition of L-type $Ca^{2+}$ channels and transient receptor potential (TRP) channels by nifedipine and 2-ABP, respectively, reduced the increase in $[Ca^{2+}]_i$ by about 50% (*Figure 1—figure supplement 2*).

## Loss of myogenic contraction affects vascular resistance and perfusion in peripheral organs

To study the consequences of a loss of the myogenic response for organ perfusion, we analyzed renal, hindlimb and cerebral perfusion (*Tan et al., 2013*; *Burke et al., 2014*). When isolated kidneys were perfused at increasing pressures, an increase in the vascular resistance could be observed, which did not differ between kidneys from wild-type mice and kidneys from Sm-q/11-KO animals (*Figure 2A*). In contrast, kidneys from Sm-12/13-KO animals showed a significantly reduced increase in renal vascular resistance (*Figure 2A*). To analyze the role of vascular smooth muscle $G_{12}/G_{13}$ and $G_q/G_{11}$ in hindlimb perfusion, we determined perfusion before and after tamoxifen-induced recombination using laser speckle imaging. Sm-12/13-KO as well as Sm-Larg-KO animals showed significantly increased hind limb perfusion after induction compared to control and Sm-q/11-KO mice (*Figure 2B and C*). Consistent with the loss of myogenic contraction in proximal cerebral arteries of Sm-12/13-KO mice, semiquantitative laser speckle imaging of brains from Sm-12/13-KO animals showed an increased brain perfusion compared to wild-type and Sm-q/11-KO animals (*Figure 2D*). This phenotype was also observed in Sm-Larg-KO animals (*Figure 2D*). These data show that the $G_{12}/G_{13}$-mediated myogenic response is required for the autoregulation of vessels of the hind limb, kidney and brain and suggests a reduced peripheral vascular resistance in these animals. Within a few weeks after induction of Sm-12/13 KO mice, we did not observe any major dysfunction of the kidney or brain in these animals when analyzing urine volume, glomerular filtration rate, blood urea nitrogen and plasma creatinine levels or after performing several tests for motoric functions and after analyzing circadian activity (*Figure 2—figure supplement 1*).

## Effect of loss of myogenic contraction on cardiac morphology and function

The consequences of defective peripheral vascular autoregulation for cardiovascular function were then analyzed by studying hearts from Sm-12/13-KO mice 3–4 weeks after tamoxifen injection. Both heart weight and cardiomyocyte cross-sectional area of Sm-12/13-KO mice were significantly increased compared to wild-type animals without any sign of fibrosis or change in capillary density (*Figure 3A*). Using magnetic resonance imaging, we confirmed that Sm-12/13-KO mice had a significantly increased left ventricular mass (*Figure 3B*) and found that left ventricular ejection fraction, stroke volume and cardiac output were significantly enhanced, whereas the heart rate was not affected (*Figure 3C*). Loss of smooth muscle $G\alpha_{12}/G\alpha_{13}$ expression had no effect on the blood pressure as reported previously (*Wirth et al., 2008*) indicating a reduced peripheral vascular resistance in these animals.

To analyze whether the myocardial hypertrophy in Sm-12/13-KO mice was accompanied by increased expression of fetal genes, which would indicate a pathological form of hypertrophy, we analyzed expression of Acta1, Myh7, Nppa and Nppb 4 weeks after induction of smooth muscle-specific $G\alpha_{12}/G\alpha_{13}$ deficiency. However, expression of none of these genes was increased (*Figure 4A*). Nppa, Nppb and Acta1 rather showed a decreased expression. In addition, expression of different fibrosis markers was unchanged, and plasma catecholamine, angiotensin II and aldosterone levels were comparable between wild-type and Sm-12/13-KO mice 4 weeks after induction (*Figure 4A and B*). Thirty five weeks after induction of smooth muscle-specific $G\alpha_{12}/G\alpha_{13}$ deficiency, hypertrophy marker genes were still unchanged or slightly reduced, and plasma levels of catecholamines,

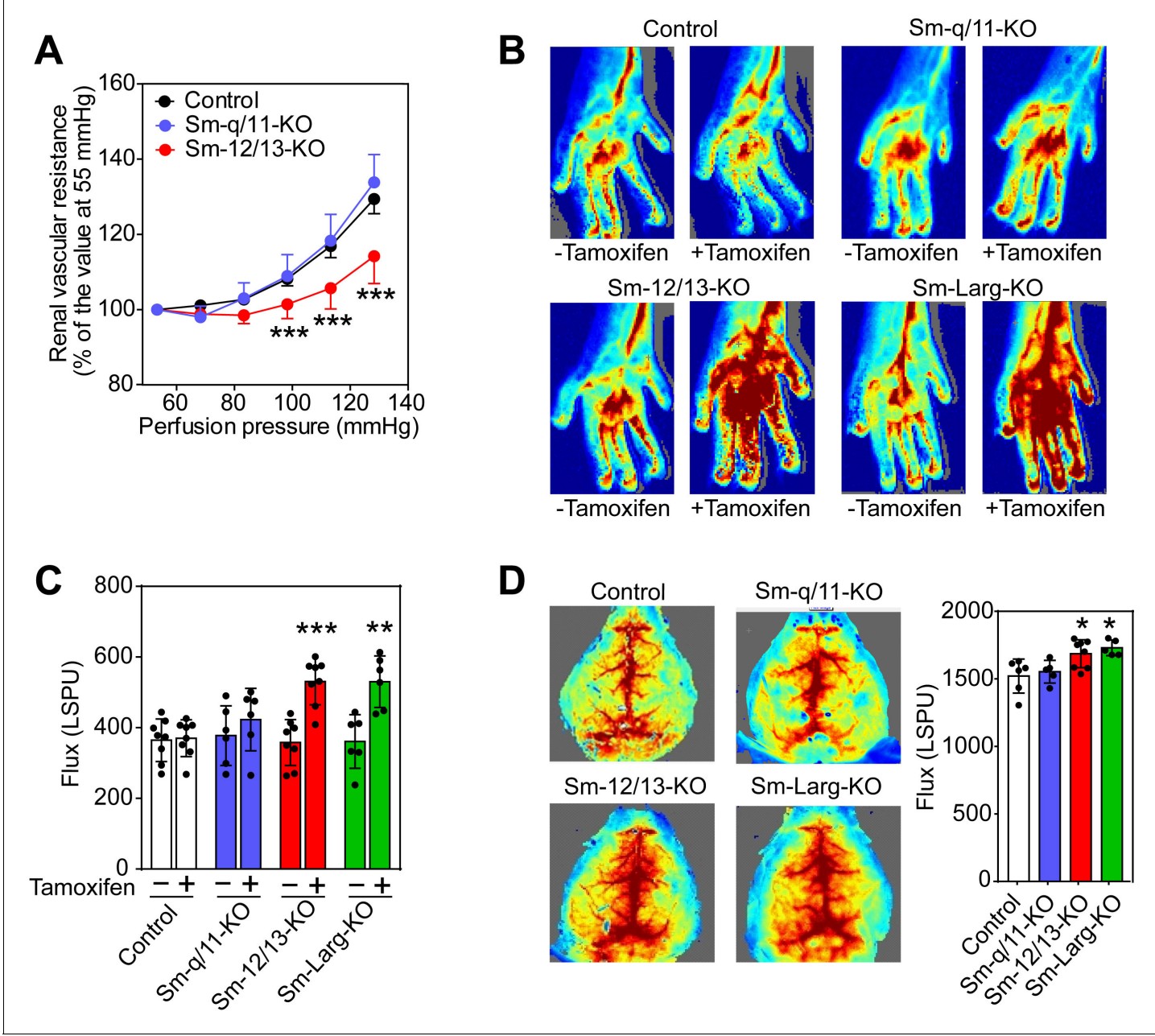

**Figure 2.** Effect of vascular smooth muscle-specific $G\alpha_{12}/G\alpha_{13}$ deficiency on vascular resistance and perfusion in peripheral organs. (A) Flow-induced increase in vascular resistance in isolated perfused kidneys of control (black circles), Sm-12/13-KO (red circles) and Sm-q/11-KO mice (blues circles) (n = 12 mice (control) and n = 7 (Sm-q/11-KO and Sm-12/13-KO)). (B, C) Laser speckle perfusion imaging of the hind limb from wild-type (control), Sm-q/11-KO, Sm-12/13-KO and Sm-Larg-KO mice before and 2 weeks after tamoxifen treatment (n = 8 mice (control and Sm-12/13-KO) and n = 6 mice (Sm-q/11-KO and Sm-Larg-KO)). (D) Laser speckle perfusion imaging of the brain from wild-type (control), Sm-q/11-KO, Sm-12/13-KO and Sm-Larg-KO mice (n = 6 mice (control), n = 5 mice (Sm-q/11-KO and Sm-Larg-KO) and n = 8 mice (Sm-12/13-KO)). Shown are representative images as well as the statistical evaluation (bar diagrams). LSPU, laser speckle perfusion units. All values are mean values ± SEM. *, $p \leq 0.05$; **, $p \leq 0.01$; ***, $p \leq 0.001$ (Bonferroni's post-hoc test (in A); unpaired t-test (in C and D; compared to control in D)).

DOI: https://doi.org/10.7554/eLife.49374.007

The following source data and figure supplement are available for figure 2:

**Source data 1.** Analysis of kidney, hind limb and brain perfusion in smooth muscle-specific $G\alpha_q/G\alpha_{11}$, $G\alpha_{12}/G\alpha_{13}$, and ARHGEF12 (LARG) deficient mice.

DOI: https://doi.org/10.7554/eLife.49374.009

**Figure supplement 1.** Effect of vascular smooth muscle-specific $G\alpha12/G\alpha13$ deficiency on Kidney and brain function.

DOI: https://doi.org/10.7554/eLife.49374.008

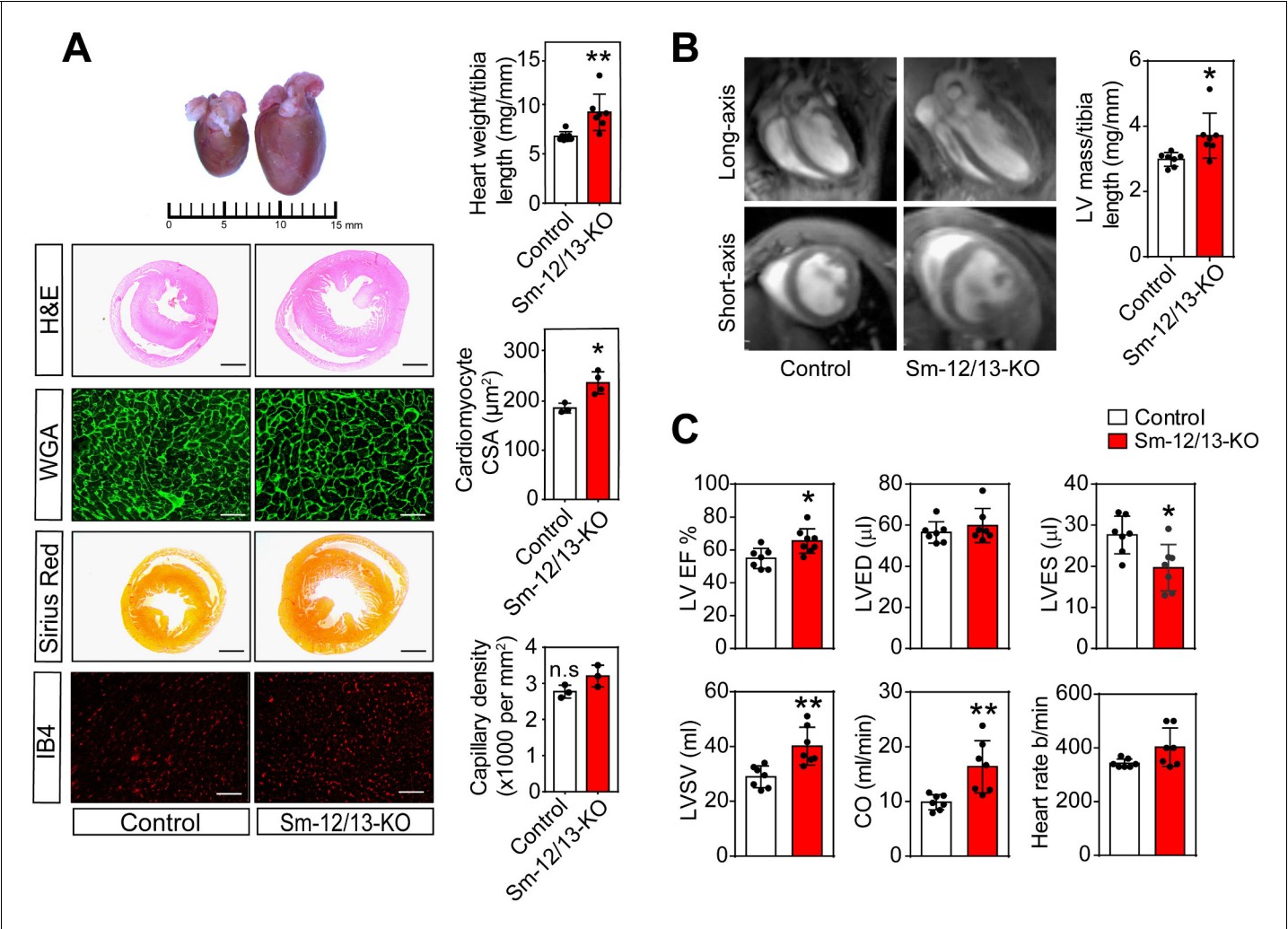

**Figure 3.** Cardiac structure and function of smooth muscle specific Gα$_{12}$/Gα$_{13}$-deficient mice. (**A**) Hearts of wild-type (control, left) and Sm-12/13-KO mice (right) were stained 4 weeks after induction with hematoxylin and eosin (H and E), wheat germ agglutinin-AF488 (WGA-AF488), picrosirius red or IB4. Quantification of heart weights (normalized to tibia length), cardiomyocyte cross-sectional area (CSA) and capillary density is represented by bar graphs. Scale bar: 2 mm (H and E- as well as Sirius red-stained sections) and 50 μm (WGA- and IB4-stained sections) (n = 7 mice per group (heart weight per tibia length), n = 4 mice per group (Cardiomyocyte area) and n = 3 mice (capillary density)). (**B**) MRI images in long-axis four chamber view and short-axis view from control and Sm-12/13-KO mice. Left ventricular mass, as calculated from the MRI data, normalized to tibia length of wild-type and Sm-12/13-KO mice is represented as bar graph (n = 7 mice per group). (**C**) MRI assessment of LV function in control (black bar) and Sm-12/13-KO mice (red bar). LVEF, left ventricular ejection fraction; LVED, left ventricular end diastolic volume; LVES, left ventricular end systolic volume; LVSV, left ventricular stroke volume; CO, cardiac output (n = 7 mice per group). All values are mean values ± SEM. *, p≤0.05; **, p≤0.01 (unpaired t-test; compared to control).

DOI: https://doi.org/10.7554/eLife.49374.010

The following source data and figure supplement are available for figure 3:

**Source data 1.** Histological and MRI analysis of hypertrophy in smooth muscle-specific Gα$_{12}$/Gα$_{13}$deficient mice.
DOI: https://doi.org/10.7554/eLife.49374.012
**Figure supplement 1.** Hearts of wild-type (control) and Sm-12/13-KO mice were stained 34 weeks after tamoxifen induction with hematoxylin and eosin (H and E), as well as picrosirius red.
DOI: https://doi.org/10.7554/eLife.49374.011

angiotensin II and aldosterone were unchanged (*Figure 4C and D*). While 2 of the six fibrosis marker genes were upregulated by 50–100% (*Figure 4C*) 35 weeks after induction, sirius red staining showed no signs of fibrosis in hearts 35 weeks after induction (*Figure 3—figure supplement 1*).

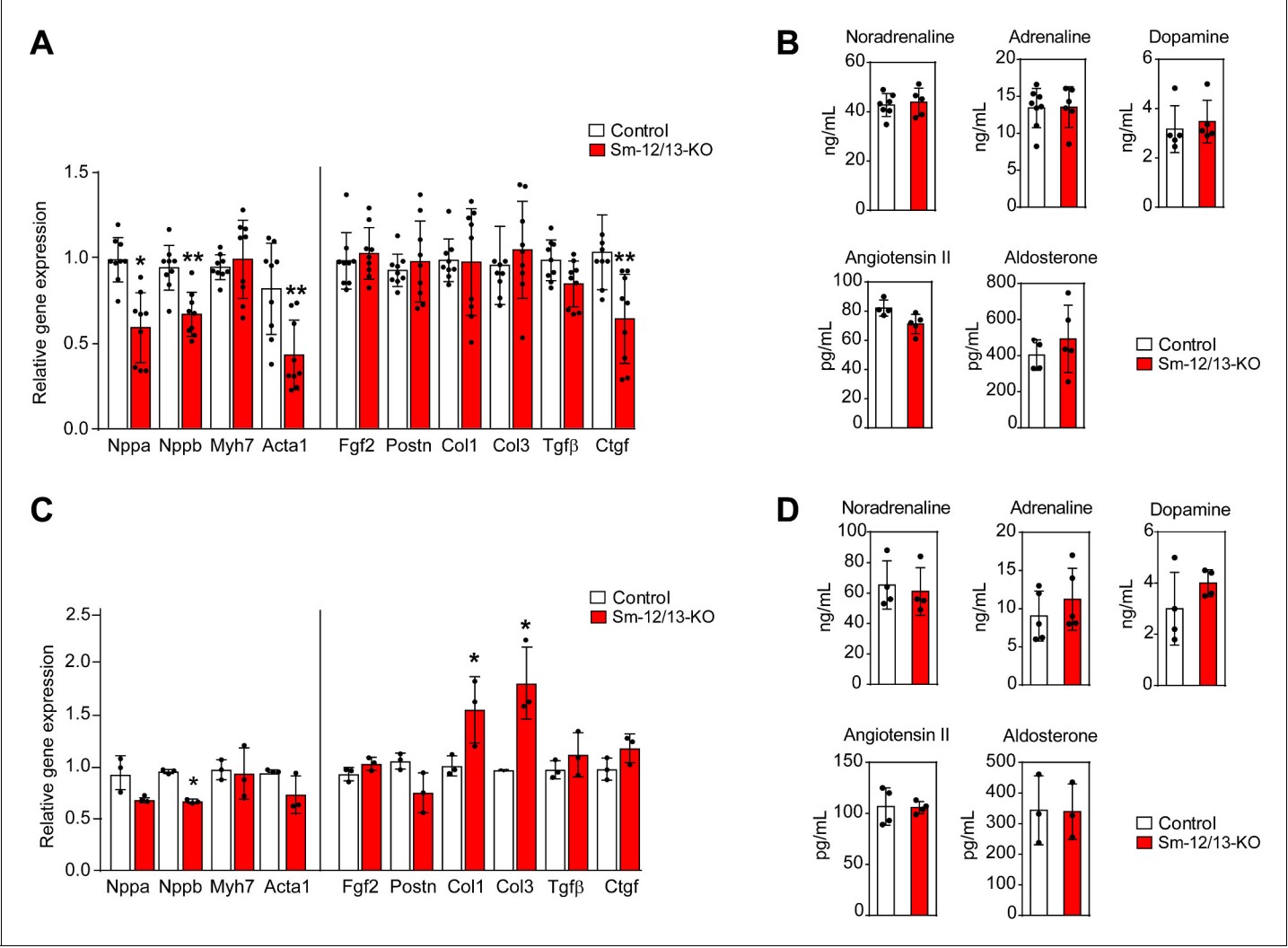

**Figure 4.** Cardiac gene expression and systemic levels of mediators in Sm-12/13-KO mice. (A and C) relative expression of myocardial genes and fibrosis marker genes in the hearts of control (white bars) and Sm-12/13-KO mice (red bars) 4 weeks (A) and 35 weeks (C) after tamoxifen treatment (n = 9 mice (A) and n = 3 mice (C)). (B and D) plasma levels of catecholamines, angiotensin-II and aldosterone in control (white bars) and Sm-12/13-KO mice (red bars) 4 weeks (B) and 35 weeks (D) after induction (n = 3–7 mice). All values are mean values ± SEM. *, p≤0.05 (compared to control; unpaired t-test).

DOI: https://doi.org/10.7554/eLife.49374.013

The following source data is available for figure 4:

**Source data 1.** Analysis of hypertrophy associated genes, catecholamines, aldosterone and angiotensin in smooth muscle-specific Gα12/Gα13deficient mice.

DOI: https://doi.org/10.7554/eLife.49374.014

## Aggravated hypotensive response to LPS in the absence of myogenic autoregulation

Since reduced vascular resistance due to an overactivity of vasodilatory stimuli including nitric oxide is believed to be a major mechanism underlying septic shock resulting in severe hypotension (*Hotchkiss et al., 2016*; *Landry and Oliver, 2001*), we tested the contribution of myogenic vasoconstriction to endotoxin-induced hypotension (*Figure 5*). We therefore determined the effect of lipopolysaccharide (LPS) on the arterial blood pressure in wild-type mice as well as in Sm-12/13-KO and Sm-Larg-KO animals. While LPS induced a transient decrease in arterial blood pressure by maximally 15 mmHg in wild-type animals, the same dose of LPS resulted in a prolonged and significantly increased hypotension when given to Sm-12/13-KO and Sm-Larg-KO mice (*Figure 5*).

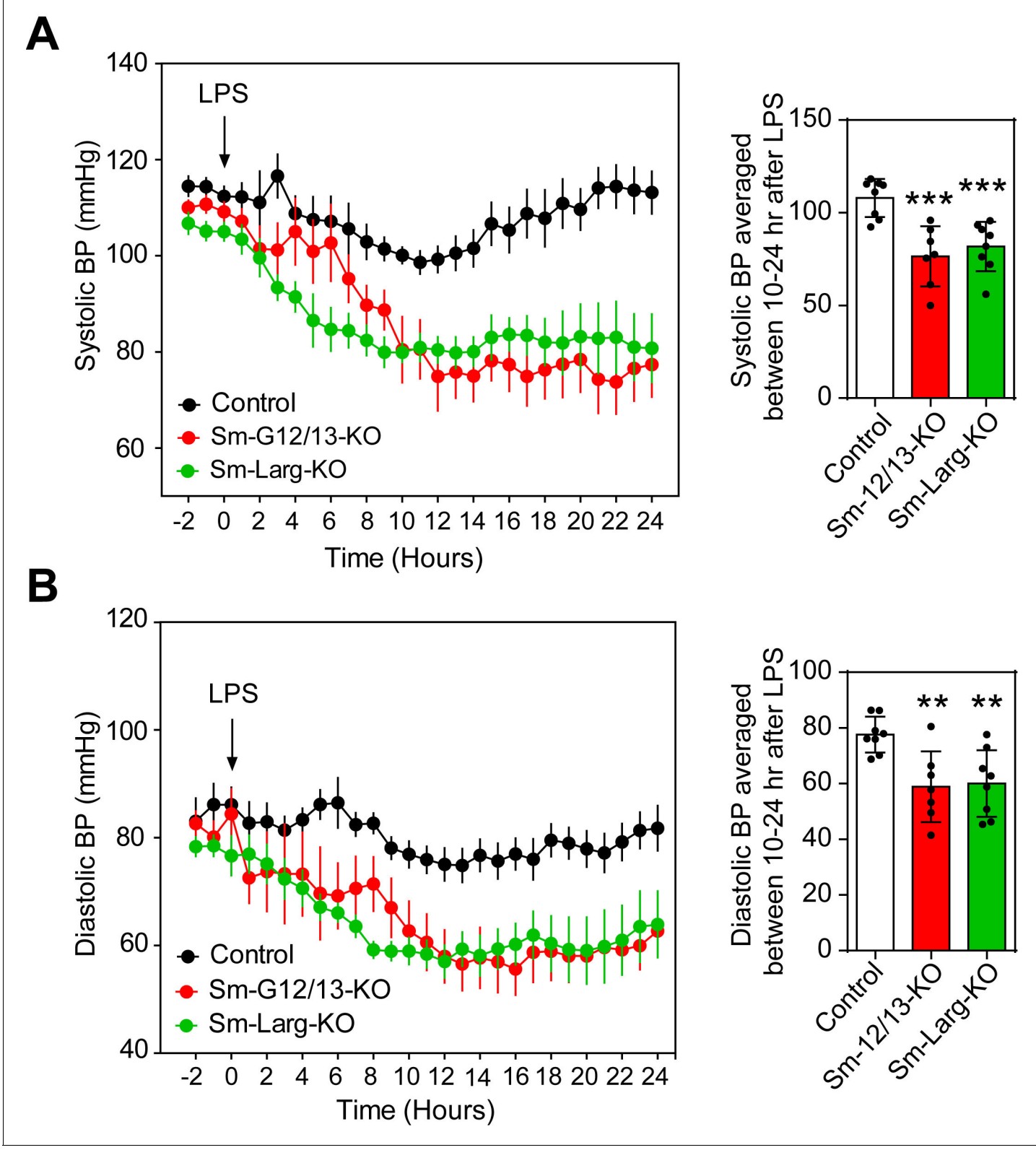

**Figure 5.** Effect of vascular smooth muscle-specific Gα₁₂/Gα₁₃ and LARG deficiency on LPS-induced hypotension. (**A** and **B**), telemetric blood pressure measurements (systolic in A, diastolic in B) were performed in wild-type (control), Sm-12/13-KO and Sm-Larg-KO mice. At the indicated time point animals were injected with 10 mg/kg of lipopolysaccharide (LPS), and the blood pressure was monitored for 24 hr (n = 8 mice (control and Sm-Larg-KO)

*Figure 5 continued on next page*

Figure 5 continued

and n = 7 mice (Sm-12/13-KO)). The bar diagrams show the statistical evaluation of the average systolic (A) and diastolic blood pressure (B) during the period between 10 and 24 hr after LPS injection. Shown are mean values ± SEM. **, p≤0.01; ***, p≤0.001 (compared to control; unpaired t-test).

DOI: https://doi.org/10.7554/eLife.49374.015

The following source data is available for figure 5:

**Source data 1.** Effects of LPS on blood pressure in $G\alpha_{12}$/$G\alpha_{13}$and ARHGEF12 (LARG) deficient mice.

DOI: https://doi.org/10.7554/eLife.49374.016

## Discussion

Small arteries are able to contract in response to increases in intravascular pressure. This myogenic vasoconstriction requires smooth muscle cell depolarization and increases in the free intracellular $Ca^{2+}$ concentration which involve opening of cation channels in the plasma membrane (*Tykocki et al., 2017*). However, multiple evidence has been provided that also GPCRs play a critical role in myogenic vasoconstriction (*Kauffenstein et al., 2012*; *Mederos Y Schnitzler et al., 2016*). Most of these vasocontractile receptors have been shown to activate both the $G_{12}$/$G_{13}$-mediated pathway resulting in activation of RhoA as well as the $G_q$/$G_{11}$-mediated pathway leading to phospholipase-β activation and $IP_3$-dependent $Ca^{2+}$ mobilization (*Gohla et al., 2000*; *Wirth et al., 2008*; *Bolz et al., 2003*; *Sauzeau et al., 2000*). This suggests that GPCR signaling through both pathways increases MLC phosphorylation and contraction by activation of MLCK through a $Ca^{2+}$-dependent pathway and by inhibition of myosin phosphatase through the Rho/Rho-kinase pathway (*Somlyo and Somlyo, 2000*; *Wirth et al., 2008*). Based on the analysis of mesenteric and cerebral arteries from smooth muscle-specific $G\alpha_q$/$G\alpha_{11}$- and $G\alpha_{12}$/$G\alpha_{13}$-deficient mice, our data, however, indicate that myogenic vasoconstriction depends on $G_{12}$/$G_{13}$-mediated signaling but not on $G_q$/$G_{11}$. This is supported by studies in vessels lacking the $G_{12}$/$G_{13}$-regulated Rho-GEF protein ARHGEF12 and is consistent with data showing that Rho/Rho-kinase-mediated signaling plays an important role in myogenic vasoconstriction (*Schubert et al., 2008*; *de Godoy and Rattan, 2011*). In contrast to our data, it has been reported that a pharmacological $G\alpha_q$/$G\alpha_{11}$ inhibitor inhibits myogenic tone in 3rd order mesenteric arteries by about 65% (*Mederos Y Schnitzler et al., 2016*; *Storch et al., 2015*). The reason for this discrepancy is not clear but may be due to compensatory effects, although we observed no defect in the myogenic response within 1 week after tamoxifen-induced loss of $G\alpha_q$/$G\alpha_{11}$ expression in smooth muscle. It can also not be excluded that the inhibitor has indirect effects through inhibition of $G_q$/$G_{11}$-mediated signaling in non-smooth muscle cells of the vessel.

GPCRs proposed to be involved in myogenic vasoconstriction comprise a heterogeneous group of receptors including purinergic $P2Y_6$, thromboxane $A_2$ (TP), sphingosine-1-phosphate, angiotensin $AT_1$ and cysteinyl leukotriene 1 ($CysLT_1$) receptors (*Mederos y Schnitzler et al., 2008*; *Schleifenbaum et al., 2014*; *Storch et al., 2015*; *Kauffenstein et al., 2016*; *Kroetsch and Bolz, 2013*), which may play different roles in myogenic vasoconstriction depending on the vascular bed and the physiological context. We have not analyzed which receptor is involved in myogenic vasoconstriction in mesenteric and cerebral arteries. However, previous studies have shown that $AT_1$ and $CysLT_1$ receptors, which appear to be activated in a ligand-independent manner are involved in myogenic vasoconstriction of mesenteric arteries (*Schleifenbaum et al., 2014*; *Storch et al., 2015*). In addition, also $P2Y_6$ receptors have been shown to be involved in the myogenic tone of mesenteric arteries, a mechanism which requires release of nucleotides acting in an autocrine or paracrine fashion through $P2Y_6$ (20). The $AT_1$ receptor plays also a role in myogenic vasocontraction of cerebral arteries (*Mederos y Schnitzler et al., 2008*).

Our observation that the increase in intracellular $Ca^{2+}$ concentration during myogenic tone development was neither affected by smooth muscle-specific $G_q$/$G_{11}$ deficiency nor by $G_{12}$/$G_{13}$ and LARG deficiency indicates that the $Ca^{2+}$-dependent contraction during myogenic tone is not mediated by GPCRs. Conflicting data have been reported as to the ability of $G_q$/$G_{11}$-coupled receptors to modulate the activity of plasma membrane ion channels required for $Ca^{2+}$ influx during myogenic tone development. While TRPC6 and TRPM4 have been reported to be activated through $G_q$/$G_{11}$-mediated signaling induced by angiotensin-II and other receptors (*Mederos y Schnitzler et al., 2008*; *Inoue et al., 2009*; *Pires et al., 2017*), another report failed to observe $G_q$/$G_{11}$-mediated

enhancement of mechanically induced TRPC currents during myogenic vasoconstriction (*Anfinogenova et al., 2011*). A recent study also suggested that TRPM4 channel opening is promoted by the Rho/Rho-kinase pathway (*Li and Brayden, 2017*) whereas other studies had found no involvement of this pathway in TRP channel regulation during myogenic tone development (*Kauffenstein et al., 2012*; *Earley and Brayden, 2015*). Our data indicate that $G_q/G_{11}$-mediated signaling in vascular smooth muscle cells does not critically contribute to increases in intracellular free $Ca^{2+}$ or smooth muscle contraction during myogenic tone development whereas the role of $G_{12}/G_{13}$-mediated signaling and its downstream RhoA-dependent signaling pathway is important for myogenic tone development but does not contribute to mechanically induced increases in intracellular $Ca^{2+}$. These data are consistent with a model of dual regulation of MLC-phosphorylation during myogenic contraction through a $Ca^{2+}$-dependent and a Rho/Rho-kinase-dependent pathway, in which $Ca^{2+}$ influx through mechanically activated plasma membrane ion channels is primarily responsible for inducing $Ca^{2+}$-dependent MLC phosphorylation, whereas activation of GPCRs via $G_{12}/G_{13}$ leads to activation of the Rho/Rho-kinase pathway (*Figure 6*). It is possible that the relevance of these two pathways differs between different vascular beds. While we observed a complete loss of myogenic vasoconstriction in mesenteric arteries from Sm-12/13-KO mice, the myogenic response in cerebral arteries was not lost but reduced by about 50%. This may be due to a more dominant role of pressure-induced depolarization and $Ca^{2+}$ signaling in cerebral arteries. $G_q/G_{11}$-mediated signaling downstream of GPCRs may contribute to $Ca^{2+}$-dependent and $Ca^{2+}$-independent signaling but is not required for myogenic vasoconstriction.

The fact that myogenic vasoconstriction of both, mesenteric and cerebral arteries, is lacking in smooth muscle-specific $G\alpha_{12}/G\alpha_{13}$-deficient mice suggests that this mouse line is a useful model to study the in vivo function of myogenic tone under physiological and pathophysiological conditions. Early observations showed that Sm-12/13-KO mice have a normal basal blood pressure but do not respond with arterial hypertension to desoxycorticosterone (DOCA)/NaCl treatment (*Wirth et al., 2008*). The present study indicates that Sm-12/13-KO mice have an increased cardiac output due to an increased stroke volume, suggesting that loss of myogenic tone in Sm-12/13-KO mice results in a reduced vascular resistance compensated by increased cardiac output. Interestingly, we did not find signs of increased sympathetic activity or changes in angiotensin II and aldosterone levels. Thus, the increased stroke volume and increased cardiac output might just be the result of a reduced afterload. Similar observations have been made during physical exercise or after pharmacological inhibition of peripheral resistance (*Andersen and Vik-Mo, 1984*; *Radovits et al., 2013*). The increased cardiac output was accompanied by myocardial hypertrophy resembling other forms of physiological hypertrophy as it did not go along with the expression of fetal genes such as those encoding natriuretic peptide A and B as well as the cardiac myosin heavy chain (Myh7) or skeletal muscle α-actin (Acta1) (*Nakamura and Sadoshima, 2018*). As described before, Acta1 expression was rather reduced 4 weeks after induction of smooth muscle-specific $G\alpha_{12}/G\alpha_{13}$ deficiency typical for physiological cardiac hypertrophy (*Nakamura and Sadoshima, 2018*; *McMullen and Jennings, 2007*). Also, 35 weeks after induction, no expression of fetal genes was observed, while 2 of 6 fibrosis marker genes showed a slight increase in expression. However, no fibrosis could be seen in histological sections of hearts 35 weeks after induction.

Our data clearly show that the reduced peripheral resistance in Sm-12/13-KO and Sm-Larg-KO mice resulting in increased cardiac output was accompanied by increased perfusion of the brain or the hind limb under basal conditions. This indicates that $G_{12}/G_{13}$-mediated myogenic tone protects organs against overperfusion. However, even 34 weeks after induction of smooth muscle-specific $G\alpha_{12}/G\alpha_{13}$ deficiency and loss of myogenic tone, we did not observe any obvious morphological or functional defects in other organs than the heart. No formation of edema was observed as would have been expected from a loss of myogenic tone. This indicates that the myogenic vasoconstriction primarily functions to establish an underlying vascular tone on which humoral or neural vasodilatory and vasocontractile stimuli act to control the vascular diameter and to also compensate to some degree alterations of myogenic tone. However, under pathological conditions when these modulatory stimuli are dysregulated such as in severe cases of hypertension or hypotension myogenic autoregulation can only partially compensate the activity of vasoactive mediators. This is also indicated by earlier observations that loss of smooth muscle $G_{12}/G_{13}$-mediated signaling had no effect on basal blood pressure but strongly suppressed mineralocorticoid hypertension in the deoxycorticosterone/salt (DOCA/salt) model (*Wirth et al., 2008*). Similarly, in the present study we show that endotoxin-

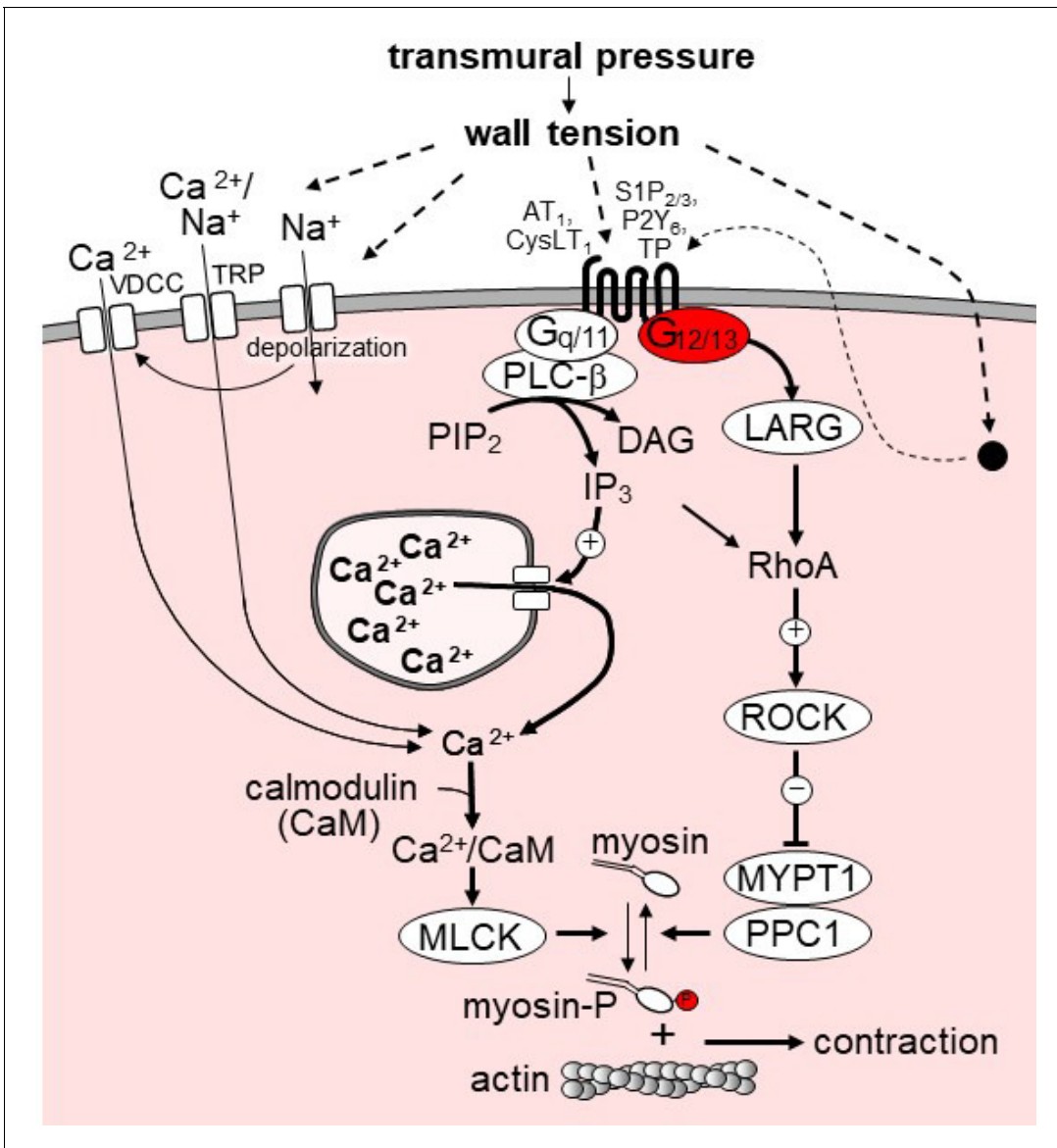

**Figure 6.** Model of the role of $G_{12}$/$G_{13}$ in vascular myogenic tone. VDCC, voltage-dependent $Ca^{2+}$ channels; TRP, transient receptor potential channels; PLC-β, phospholipase C-β; PIP$_2$, phosphatidyl inositol bisphosphate; DAG, diacyl glycerol, IP$_3$, inositol-1,4,5-triphosphate; LARG, RhoGEF protein ARHGEF12; MYPT1 and PPC1, regulatory and catalytic subunit of myosin phosphatase, respectively; ROCK, Rho-kinase. ROCK phosphorylates MYPT1 and thereby inhibits myosin phosphatase activity. AT$_1$, angiotensin AT$_1$ receptor; CysLT$_1$, cysteinyl leukotriene receptor 1; S1P$_{2/3}$, sphingosine-1-phosphate receptors 2 and 3; P2Y$_6$, purinergic receptor Y6; TP, thromboxane A$_2$ receptor. AT$_1$ and CysLT$_1$ have been shown to be activated by increased vascular pressure in a ligand independent manner, whereas activation of S1P$_{2/3}$, TP and P2Y$_6$ is believed to require formation or release of the respective receptor ligand. For details see text.

DOI: https://doi.org/10.7554/eLife.49374.017

induced hypotension is aggravated in Sm-12/13-KO mice suggesting that under septic conditions, which result in strong vasodilatation and hypotension, $G_{12}$/$G_{13}$-mediated myogenic tone is still intact and provides partial resistance against vasodilatory influences.

In conclusion, our findings indicate that $G_{12}$/$G_{13}$-mediated signaling through Rho and Rho-kinase in vascular smooth muscle plays a key role in mediating myogenic vasoconstriction, whereas $G_q$/$G_{11}$-mediated signaling plays no or only a minor role and appears not to be required for $Ca^{2+}$-mediated signaling in myogenic tone development. Our data suggest that $Ca^{2+}$-dependent signaling leading to myogenic contraction mainly relies on $Ca^{2+}$ influx through plasma membrane ion channels, whereas the calcium-independent Rho/Rho-kinase-mediated signaling is induced through $G_{12}$/$G_{13}$-

coupled receptors of which several have been reported to be involved in myogenic vasocontraction. Our data also indicate that myogenic vasoconstriction is required to maintain vascular tone as well as local and systemic vascular resistance under normal and disease conditions.

# Materials and methods

**Key resources table**

| Reagent type (species) or resource | Designation | Source or reference | Identifiers | Additional information |
|---|---|---|---|---|
| Genetic reagent (*Mus musculus*) | *Gna12*$^{-/-}$ | PMID: 12077299 | RRID:MGI:3819345 | |
| Genetic reagent (*Mus musculus*) | *Gna13*$^{flox/flox}$ | PMID: 14528298 | RRID:MGI:3819345 | |
| Genetic reagent (*Mus musculus*) | *Gnaq*$^{flox/flox}$ | PMID: 11689889 | RRID:MGI:3819271 | |
| Genetic reagent (*Mus musculus*) | Gna11$^{-/-}$ | PMID: 9687499 | RRID:MGI:3819271 | |
| Genetic reagent (*Mus musculus*) | *Larg*$^{flox/flox}$ | PMID: 18084302 | RRID:MGI:3819344 | |
| Genetic reagent (*Mus musculus*) | *Myh11*-CreERT2 | PMID: 18084302 | RRID:IMSR_JAX:019079 | |
| Antibody | anti-phospho-MYPT1 (rabbit polyclonal) | Merck Millipore | Catalog: 36–003 RRID:AB_310812 | WB (1:1000) |
| Antibody | anti-MYPT1 (rabbit polyclonal) | Cell Signalling Technologies | Catalog: 2634 RRID:AB_915965 | WB (1:500) |
| Antibody | anti-GAPDH (rabbit monoclonal) | Cell Signalling Technologies | Catalog: 2118 RRID:AB_561053 | WB (1:1000) |
| Commercial assay or kit | Rho-GLISA | Cytoskeleton Inc | BK124 | for determination of Rho activity |
| Ccommercial assay or kit | 3-CAT ELISA-ImmuSmol | LDN | BA E-5600 | for determination of adrenaline, dopamine and noradrenaline levels |
| Commercial assay or kit | Angiotensin II ELISA kit | ENZO Life Sciences | ADI-900–204 | |
| Commercial assay or kit | Aldosterone ELISA kit | ENZO Life Sciences | ADI-900–173 | |
| Chemical compound, drug | Y-27632 | Sigma Aldrich | Y0503 | 10 µM |
| Chemical compound, drug | 2-Aminoethyl diphenylborinate (2-ABP) | Sigma Aldrich | D9754 | 1 µM |
| Chemical compound, drug | Nifedipine | Sigma Aldrich | N7634 | 1 µM |
| Chemical compound, drug | Fura-2AM | Thermo Fisher Scientific | F1225 | 12.5 µM |
| Chemical compound, drug | Ionomycin | Sigma Aldrich | I0634 | 1 µM |
| Chemical compound, drug | LPS (Lipopolysaccharide) | Sigma Aldrich | L2630 | 10 mg/kg b.w |
| Other | WGA-AF488 | Thermo Fisher Scientific | W11261 | Histology (20 µg/ml) |
| Other | IB4-AF488 | Thermo Fisher Scientific | I21411 | Histology (1:200) |

## Solutions and agents

Tamoxifen, lipopolysaccharide (LPS), 2-aminoethoxydiphenyl borate (2-ABP), nifedipine, phenylephrine, acetylcholine, Y-27632 and ionomycin were purchased from Sigma-Aldrich. U-46619 was from Cayman Europe and Fura-2-AM was purchased from ThermoFischer Scientific. Krebs-Ringer bicarbonate-buffered salt solution (KRB) contained (in mmol/l): 118.5 NaCl, 4.7 KCl, 2.5 CaCl$_2$, 1.2 MgSO$_4$, 1.2 KH$_2$PO$_4$, 25.0 NaHCO$_3$ and 5.5 glucose. The KRB solution was continuously aerated with 95% O$_2$/5% CO$_2$ and maintained at 37°C.

## Genetic mouse models

The generation of floxed alleles of the genes encoding G$\alpha_q$ (*Gnaq*), G$\alpha_{13}$ (*Gna13*) and LARG (*Arhgef12*) as well as of null alleles of the genes encoding G$\alpha_{11}$ (*Gna11*) and G$\alpha_{12}$ (*Gna12*), has been described previously (*Wirth et al., 2008*; *Moers et al., 2003*; *Wettschureck et al., 2001*). All mouse lines were on a C57/BL6 background or have been backcrossed at least ten times on this genetic background. The inducible, smooth muscle-specific Cre transgenic mouse line was described before (*Wirth et al., 2008*). Since this line carries the transgene on the Y chromosome, all experiments were performed in males. To induce Cre-mediated recombination, 8- to 10-week-old mice were injected intraperitoneally with tamoxifen (1 mg/day) for five consecutive days as described before (*Wirth et al., 2008*). Mice were housed in standard cages at constant room temperature and humidity on a 12 hr light/dark cycle and had free access to standard chow pellets and tap water. All animal care and use procedures in this study were approved by the local authorities (Regierungspräsidia Karlsruhe and Darmstadt).

## Pressure myography

Pressure myograph experiments were performed as described previously (*Wang et al., 2016*; *Wang et al., 2015*). In brief, 10–14 days after tamoxifen injection, first order and second order mesenteric arteries were removed from the mesentery and were mounted between two glass micropipettes seated in a pressure myograph chamber (Danish Myo Technology; 110P and 114P). The external diameter of the artery was visualized and recorded with a CCD camera using MyoView software. Arterial segments were pressurized step-wise from 20 mmHg to 120 or 140 mmHg as indicated. Myogenic tone was expressed as the percentage of passive diameter ((passive diameter - active diameter)/passive diameter x 100). Proximal cerebral arteries were carefully isolated and similarly proceeded on a video-monitored arteriograph system (Living System; LSI, Burlington, VT) which allowed to measure changes in arterial internal diameter of cannulated cerebral arteries following stepwise increases in intraluminal pressure from 10 to 100 mmHg.

## Determination of [Ca$^{2+}$]$_i$

For measurements of the free intracellular Ca$^{2+}$ concentration arterial segments were mounted in a pressure myograpgh chamber (110P) and incubated with 6 ml of Krebs solution containing 12.5 μM Fura-2-AM, and 0.02% pluronic F-12. The chamber was connected to a pressure myograph unit and left at 37°C for 2 hr in the dark while under constant aeration with 95% O$_2$/5% CO$_2$. At the end of the loading period, the artery was washed two times with Krebs buffer. The arterial segment was then pressurized step-wise (20 mmHg) from 20 to 140 mmHg. During the pressure steps, the artery was illuminated with light of 340 and 380 nm wavelengths using an Olympus CellR MT-20 mercury-xenon burner connected to Olympus (IX71) inverted microscope. The emitted fluorescence (340/380 nm ratio) was assessed by CellR software. At the end of the experiment all arterial segments were treated with 1 μM ionomycin to induce the maximal calcium level. The fluorescence ratio (340/380 nm) for each pressure step was expressed as percent of the maximal fluorescent ratio observed in the presence of ionomycin.

## Rho-activation assay

RhoA activity was determined using the G-LISA RhoA activation assay biochemical kit (Cytoskeleton, Inc). Active RhoA was assessed according to the manufacturer's instructions. In brief, first and second order mesenteric arteries were isolated and mounted in a pressure myograph. The pressure was increased step-wise up to 80 mmHg until the myogenic contraction was apparent. In case of Sm-12/13-KO and Sm-Larg-KO animals in which no myogenic contraction could be observed, arterial

segments were pressurized step-wise up to 80 mmHg and RhoA activity was determined after 5 min. Arteries were immediately snap-frozen and lyzed and processed according to the manufacturer's instructions. 50 µl of the protein sample (0.5 µg/µl) from arterial segments were added to 96-well plates coated with the Rho binding domain of Rho effector protein and incubated at 4°C for 30 min under shaking. The plates were subsequently incubated with anti-RhoA antibody and secondary horseradish peroxidase-conjugated antibody for 45 min at room temperature. Active RhoA levels were determined by measuring absorbance at 490 nm using a microplate spectrophotometer.

## Analysis of MYPT1 phosphorylation

Whole mesenteric arterial beds including first to fourth order vessels were isolated in a cold Krebs buffer. Open ends of the arteries were closed with sutures, the superior mesenteric artery was cannulated in the pressure myograph chamber (P1 side) and vessels were incubated for 20 min at 37 C. The intraluminal pressure was increased stepwise to 80 mmHg using a pressure myograph unit (Danish Myo Technology). After reaching the target pressure, arteries were snap frozen and transferred to a vial with trichloroacetic acid in acetone (10% wt/vol) as described (*Kitazawa et al., 2003*) and stored at −80°C. Frozen arteries were washed in acetone, homogenized and lysed in a radioimmunoprecipitation assay (RIPA) buffer containing 150 mM NaCl, 50 mM Tris-HCl (pH 7.4), 5 mM EDTA, 0.1% (wt/vol) SDS, 0.5% sodium deoxycholate, and 1% Triton X-100 as well as protease inhibitors (10 mg/ml leupeptin, pepstatin A, 4-(2-aminoethyl) benzenesulfonyl-fluoride, and aprotinin) and phosphatase inhibitors (PhosSTOP, Roche). Total tissue lysates were separated by SDS-PAGE. Proteins were then transferred onto nitrocellulose membranes followed by blocking for 1 hr with TBST containing 5% skim milk and 1% BSA. Membranes were incubated overnight with primary antibodies, for additional 2 hr with HRP-conjugated secondary antibodies (Cell Signaling Technology) at room temperature and were then developed using the ECL detection system (Thermo Scientific Pierce, Life Technologies). Intensity values of bands representing phosphorylated sites of proteins were normalized to the intensity of the band representing total protein. Antibody directed against phosphorylated MYPT1 was from Merck-Millipore (36-003) and antibodies against MYPT1 (catalog 2634) and GAPDH (catalog 2118) were obtained from Cell Signalling Technologies.

## Kidney perfusion experiments

Isolated kidneys were perfused ex-situ as described in detail previously (*Demerath et al., 2014*). The perfusion medium consisted of a modified Krebs-Henseleit buffer supplemented with bovine serum albumin (6 g/100 ml) and human erythrocytes (10% hematocrit). The renal vein was cannulated and samples of the venous perfusate were collected every 2 min to determine the renal blood flow. Three samples were taken during each experimental period and the last two values of each experimental period were averaged for statistical analysis. To determine the pressure-dependent regulation of vascular resistance, the perfusion pressure was changed in steps of 15 mmHg between 130 mmHg and 55 mmHg.

## Laser speckle perfusion measurements

Hind limb and cerebral blood flow was analyzed using a laser speckle contrast images (MoorFLPI-2, Moor Instruments, UK) mounted on an adjustable tripod and connected to a standard laptop computer equipped with real-time data acquisition software (MoorFLPI measurement software, Moor Instruments). Video frame rates of flow within the microcirculation are provided at up to one image per second at a maximum resolution of 49,000 pixels/cm$^2$ (for reference, the system simultaneously records a corresponding gray scale image with an integrated charge-coupled device video camera). All experiments were performed under 1.5% isoflurane anesthesia. The whole equipment was placed on an anti-vibration table. For the analysis of cerebral blood flow, the hair was removed from the cranium, and the head was fixed in a stereotaxic frame (Stoelting, USA) with a self-controlled heating system keeping temperature at 37°C. To enhance the image quality by minimizing the effects of static scattering elements, mineral oil was placed on the surface of the brain (*Boas and Dunn, 2010*). After 10 min, live brain images were sampled at a sampling rate of 1 Hz for flux measurements. A region of interest was defined (flexible in size and location), and the mean flow in that region was calculated and recorded in real time. Laser speckle contrast imager data were evaluated using MoorFLPI review software (moorFLPI Full-Field Laser Perfusion Imager Review Version 4.0). A

mean image was calculated from the 60 perfusion images taken during a 5 min time span. Mean flux values were determined from the whole brain region. Hind limb perfusion was evaluated as described before (*Ungerleider et al., 2016*).

## Heart MRI measurements

To assess the morphological and functional changes, LV structure and function were determined by cardiac magnetic resonance imaging (MRI) after 3 weeks of tamoxifen treatment. Isoflurane (2.0% v/v) anesthesia was delivered to mice in an oxygen/medical air (0.5/0.5 L/min) mixture during the measurement. The body temperature was maintained 37˚C by a thermostatically regulated water flow system during the entire imaging protocol. Measurements were performed on a 7.0 T Bruker PharmaScan, equipped with a 760 mT/m gradient system using a $^1$H Array MRI Cryoprobe and the Intra-Gate self-gating tool (Bruker, Ettlingen, Germany). The measurement is based on the gradient echo method (repetition time = 6.2 ms; echo time = 1.3 ms; field of view = 2.20 × 2.20 cm; slice thickness = 1.0 mm; matrix = 128 × 128; repetitions = 100; resolution 0.0172 cm/pixel). The imaging plane was localized using scout images showing the 2- and 4-chamber view of the heart, followed by acquisition in short axis view, orthogonal on the septum in both scouts. Multiple contiguous short-axis slices consisting of 9 or 10 slices were acquired for complete coverage of the left ventricle. MRI data were analysed using Qmass digital imaging software (Medis, Leiden, Netherlands).

## Telemetric blood pressure measurement and LPS treatment

Mice were implanted with a pressure sensing transmitter (PA-C10, Data Sciences International) as described previously (*Wirth et al., 2008*) to measure the blood pressure in conscious animals. After a recovery period of 10 days, basal blood pressure was recorded for 3 days and Cre-recombination was induced by injection of tamoxifen as described above. One week after induction of Cre-recombination, LPS (10 mg / kg body weight, dissolved in PBS) was administered through intraperitoneal injection. Blood pressure, temperature and various other parameters (*Shrum et al., 2014*) were monitored.

## Kidney function tests

For urine volume measurements, mice were separated in metabolic cages with free access to water and powdered food pellets 2 weeks after tamoxifen induction, and urine was collected during 24 hr. The glomerular filtration rate (GFR) was calculated from FITC-sinistrin plasma clearance in conscious mice as described (*Schreiber et al., 2012*). To evaluate blood urea nitrogen (BUN) and creatinine, blood was collected after two weeks of tamoxifen induction, and plasma samples were analyzed by a diagnostic laboratory (IDEEX, Germany).

## Motor coordination and circadian activity tests

Experiments were performed 3 weeks after tamoxifen induction. For the rotarod test, mice were trained for 3 days on a rotarod (Panlab, Barcelona, Spain). After a training session, mice were subjected to test session with accelerating speeds (4–40 rpm). Each test session was composed of 2 trials on the rotarod with a maximum duration of 5 min, and an inter-trial interval of 15 min. The retention time on the rotarod for each mouse was measured. For the cage grip test, mice were placed on a metal cage lid and lifted approximately 10 cm from the ground. Slowly cage lids were tilted (180 degree), and the latency of the mouse to release the cage lid was measured. All mice were trained for two days prior to the experiment. For bar cross assays, animals were placed on a metal rod with a diameter of 3 cm and a length of 60 cm fixed by a G-clamp to a table. The hindpaw lateral slippings were observed during a 5 min test period. To determine circadian activity, 10 weeks old mice were single housed in a Phenomaster system (TSE-Systems) with free access to food and water. After 24 hr of adaptation, activity was determined by light (laser beam) barrier interruption, and data were analyzed using the PhenoMaster-Software (TSE-Systems, Bad Homburg, Germany).

## Histology

Freshly dissected hearts were perfused and fixed in 4% paraformaldehyde and embedded in paraffin after dehydration. Embedded sections (10 µm) were stained with hematoxylin and eosin as well as with picrosirius red according to standard protocols. From the stained heart sections, 6–10 randomly

chosen frames were quantified to assess the degree of heart fibrosis using ImageJ software. To quantify the myocyte cross-sectional area, heart sections were incubated with DAPI (1:1000) and WGA-AF488 conjugate (20 µg/ml in PBS) for 2 hr. Cardiomyocyte cross-sectional area (CSA) was quantified in 10 randomly chosen frames from the stained sections using ImageJ software. To assess the capillary density, heart sections were incubated with DAPI (1:1000) and IB4-AF488 conjugate (1:200) in PBS overnight and quantified using ImageJ software.

### Determination of catecholamine, angiotensin II and aldosterone levels

Two weeks after tamoxifen treatment, blood was collected from different experimental groups and plasma was prepared and snap-frozen. Plasma samples were stored at −80°C. To assess dopamine, noradrenaline and adrenaline concentrations 3 Catecholamines ELISA kit (BA-E-5600) from ImmuS-mol was used. Plasma angiotensin II and aldosterone levels were determined using an ELISA from ENZO Life Sciences (ADI-900–204 and ADI-900–173). Analyses were performed according to the manufacturer's instructions.

### qRT-PCR

Quantitative RT-PCR was performed as described (*Sivaraj et al., 2013*). Hearts were collected 4 weeks and 35 weeks after tamoxifen induction. RNA was isolated using Qiagen RNeasy mini kit, and cDNA was preamplified using ProtoScript II Reverse Transcriptase (M0368S, New England BioLabs). Primers were designed with the online tool provided by Roche, and quantification was performed using the LightCycler 480 Probe Master System (Roche). Relative expression levels were obtained by normalization with *GAPDH*. Primer sequences were as follows: ANP, forward: 5'-cacagatctgatggatttcaaga-3', reverse: 5'-cctcatcttctaccggcatc-3'; BNP, forward: 5'-gtcagtcgtttgggctgtaac-3', reverse: 5'-ggaaagagacccaggcaga-3'; Myh7, forward: 5'-cgcatcaaggagctcacc-3', reverse: 5'-ctgcagccgcagtaggtt-3'; Acta1, forward: 5'-tgaagcctcacttcctaccc-3', reverse: 5'-cgtcgcacatggtgtctagt-3', PGC-1α, forward: 5'-tgaaagggccaaacagagag-3', reverse: 5'-gtaaatcacacggcgctctt-3'; Tgfβ, forward: 5'-TGGAGCAACATGTGGAACTC-3', reverse: 5'-CAGCAGCCGGTTACCAAG-3'; Fgf2, forward: 5'-CGGCTCTACTGCAAGAACG-3', reverse: 5'-TGCTTGGAGTTGTAGTTTGACG-3'; Ctgf, forward: 5'-TGACCTGGAGGAAAACATTAAGA-3', reverse: 5'-AGCCCTGTATGTCTTCACACTG-3'; Postn, forward: 5'-CGGGAAGAACGAATCATTACA-3', reverse: 5'-ACCTTGGAGACCTCTTTTTGC-3'; Col1, forward: 5'-CATGTTCAGCTTTGTGGACCT-3', reverse: 5'-GCAGCTGACTTCAGGGATGT-3'; Col3, forward: 5'-TCCCCTGGAATCTGTGAATC-3', reverse: 5'-TGAGTCGAATTGGGGAGAAT-3' Gapdh, forward: 5'-AGCTTGTCATCAACGGGAAG-3', reverse: 5'-TTTGATGTTAGTGGGGTCTCG-3'.

### Statistical analysis

Statistical data analysis is included in every figure and described in detail on the respective figure legends. The exact *P* values and numbers of number of independent experiments (*n*) are stated in the figures and figure legends. Trial experiments or experiments done previously were used to determine sample size with adequate statistical power. Samples were excluded in cases where RNA/cDNA quality or tissue quality after processing was poor (below commonly accepted standards). Animals were excluded from experiments if they showed any signs of sickness. The investigator was blinded to the group allocation and during the experiment. Data are presented as means ± SEM. All statistical analyses were performed using Prism five software (GraphPad). A level of p<0.05 was considered significant and reported to the graphs. Comparisons between two groups were performed using unpaired 2-tailed Student's *t*-test and multiple group comparisons were performed by ANOVA followed by Bonferroni's post-hoc test.

## Acknowledgements

The authors wish to thank Svea Hümmer for secretarial help and Kathrin Heil and Dagmar Magalei for technical support. This work was supported by the German Centre for Cardiovascular Research and the Max Planck society.

## Additional information

### Funding

| Funder | Grant reference number | Author |
|---|---|---|
| Max-Planck-Gesellschaft | Open-access funding | Stefan Offermanns |

This study was funded by the Max Planck Society.

### Author contributions

Ramesh Chennupati, Data curation, Software, Formal analysis, Validation, Investigation, Visualization, Methodology, Writing—review and editing; Angela Wirth, Conceptualization, Investigation; Julie Favre, Formal analysis, Validation, Methodology; Rui Li, Astrid Wietelmann, Frank Schweda, Methodology; Rémy Bonnavion, Young-June Jin, Investigation, Methodology; Nina Wettschureck, Writing—review and editing; Daniel Henrion, Validation, Methodology, Writing— review and editing; Stefan Offermanns, Conceptualization, Supervision, Funding acquisition, Writing—original draft, Writing—review and editing

### Author ORCIDs

Ramesh Chennupati (iD) https://orcid.org/0000-0002-3994-5285
Stefan Offermanns (iD) https://orcid.org/0000-0001-8676-6805

### Ethics

Animal experimentation: All animal care and use procedures in this study were approved by the local authorities (protocol numbers: B2-1031, B2-1166, B2-1069 Regierungspräsidia Karlsruhe and Darmstadt).

### Decision letter and Author response

Decision letter https://doi.org/10.7554/eLife.49374.020
Author response https://doi.org/10.7554/eLife.49374.021

## Additional files

### Supplementary files

• Transparent reporting form
DOI: https://doi.org/10.7554/eLife.49374.018

### Data availability

All data generated or analysed during this study are included in the manuscript and supporting files. Source data files have been provided for Figures 1–5.

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
