## [Decision Letter]

Thank you for submitting your article "Myogenic vasoconstriction requires G_12_/G_13_ and Arhgef12 to maintain local and systemic vascular resistance" for consideration by *eLife*. Your article has been reviewed by three peer reviewers, including Mark T Nelson as the Reviewing Editor and Reviewer #1, and the evaluation has been overseen by Harry Dietz as the Senior Editor.

The reviewers have discussed the reviews with one another and the Reviewing Editor has drafted this decision to help you prepare a revised submission.

Summary:

In this manuscript, Chennupati et al. investigate the role of G_12/13_ and Gq/11 signaling pathways downstream of G-protein-coupled receptors (GPCRs) in myogenic constriction of resistance arteries (First/Second-order mesenteric, cerebral). Using mice with an inducible, smooth muscle-specific deficiency of Gαq/11 or Gα_12/13_, they found that G_12/13_ signaling plays a major role in myogenic constriction, whereas Gq/11 signaling does not. These results are somewhat surprising given numerous literature reports that Gαq/11-coupled GPCRs act as mechanosensors in the myogenic response, sharpening the impact of the manuscript. A role for the ARHGEF12 (Rho guanine nucleotide exchange factor) and RhoA-mediated signaling in Gα_12/13_-dependent myogenic responses was confirmed using inducible, smooth muscle-specific Arhgef12-deficient animals and biochemical (RhoA kinase activity measurements) and pharmacological (Rho-kinase inhibitor Y-27632) approaches. Chennupati et al. further demonstrated that the reduced systemic vascular resistance associated with the loss of myogenic constriction in Gα_12/13_-deficient mice was accompanied by a compensatory increase in cardiac output and physiological cardiac hypertrophy, with the net result being an increase in the perfusion of peripheral organs. The experimental approaches used are sound and compellingly support the authors' conclusion that G_12/13_- and RhoA-mediated signaling plays a key role in myogenic vasoconstriction and that myogenic tone is required to maintain local and systemic vascular resistance.

Essential revisions:

1) Why first/second-order mesenteric arteries? A subset of experiments in third order mesenteric arteries should be done (or shown, if already performed), given the previous report (cited by the authors) that inhibition of Gq/11 signaling blunts myogenic tone in third order mesenteric arteries.

2) There appears to be at least two mechano-sensitive processes: 1) Membrane depolarization and opening of voltage-dependent calcium channels. 2) G_12/13_- and RhoA-mediated decrease in MLC dephosphorylation. The authors suggest that based on one supplementary figure without original data that neither Gαq/11 or Gα_12/13_ is involved in the pressure-induced increase in calcium in larger mesenteric arteries, which require relatively high pressures for a myogenic response. This would be a significant result on its own if there were more data. Furthermore, what is swept under the rug, is that the pressure-induced constrictions of cerebral arteries are only attenuated in the Gα_12/13_ KO, and there are no accompanying calcium or membrane potential measurements. Possible explanations should be discussed, e.g. more robust engagement of pressure-induced depolarization/calcium elevation in cerebral arteries. This should be discussed. It is possible that a balance between two mechanisms may determine the sensitivity to pressure.

3) Image of brain perfusion in SM-q/11-KO mice does not appear to be representative of summary data (overall impression of high intensity areas is closer to that of the other KOs than it is to the control). Should replace image with one that is more representative and provide additional detail on how images were quantified.

4) G_12/13_ would presumably need to be activated in order to signal to RhoA and downstream response to inducers of myogenic constriction. It is not clear how this pathway is turned on. Ex vivo the authors use increasing vascular pressure. While it is possible that there is mechanical input the authors state the more likely physiological mechanism in their conclusion, indicating that "RhoA/Rho kinase mediated signaling is induced through G_12/13_ coupled receptors". The schematic also shows black dots which are unlabelled but perhaps meant to connote some GPCR ligand? The authors should clarify what they think is happening ex vivo, as well as in vivo where absence of G_12/13_ (thus presumably loss of signaling through a GPCR coupled to it) leads to changes in perfusion and ultimately alterations in cardiac function. The literature, or the authors own findings, using KOs for G_12/13_ linked receptors such as S1P, TXA2, PAR1 should be reviewed and used to provide a more scholarly discussion of myogenic vascular control mechanisms.

5) The schematic in Figure 5 leaves out some of the molecules involved in the transduction pathways under study. In light of the particular focus on the RhoA/Rho Kinase pathway, the myosin binding subunit MYPT-1 which is not depicted is the known substrate of Rho kinase and it has been widely used as a readout for activation of this pathway. Phospor-Abs are available. Evidence that this substrate is regulated – both increased in response to transmural pressure and lost in the G_12/13_ knockout, should be provided. Measurement of myosin phosphorylation is also possible but less useful since it reflects both the Ca increase in MLCK and Rho-mediated decrease in MLC phosphatase. The schematic in Figure 5 could be enhanced.

6) The Myh7, Acta1, and other fibrosis-related genes were not increased in the hypertrophied heart of the Sm-12/13 mice. How do cardiomyocytes become hypertrophic without an increase of contractile proteins? The authors should discuss this point.

7) What are the consequences of hyper-perfusion of blood in kidney and brain? Is urine volume comparable between the Sm-12/13 and control mice? Did the authors check the sleep-wakefulness? Although the functional PET imaging is unnecessary, those regular physiological examinations might be done by the authors?

8) The main target of RhoA is likely to be Rho-kinase in the regulation of vascular resistance in the present study; however, there has no obvious data in the present study. Therefore, the authors should tone down the description in the Discussion section.

9) Can the authors overexpress constitutive active RhoA in the vascular smooth muscle in the Sm-12/13 KO to see whether high output is changed or unchanged by the vascular tones? (this is not a mandatory experiment).

---

## [Author Response]

Essential revisions:1) Why first/second-order mesenteric arteries? A subset of experiments in third order mesenteric arteries should be done (or shown, if already performed), given the previous report (cited by the authors) that inhibition of Gq/11 signaling blunts myogenic tone in third order mesenteric arteries.

We had performed a series of experiments also in third order mesenteric arteries and basically found the same as in first and second order mesenteric arteries, namely a strong reduction of myogenic vasoconstriction in vessels from Sm-12/13-KO mice and no defect in vessels from Sm-q/11-KO animals. These data are now presented in Figure 1—figure supplement 1E-G.

2) There appears to be at least two mechano-sensitive processes: 1) Membrane depolarization and opening of voltage-dependent calcium channels. 2) G_12/13_- and RhoA-mediated decrease in MLC dephosphorylation. The authors suggest that based on one supplementary figure without original data that neither Gαq/11 or Gα_12_/_13_ is involved in the pressure-induced increase in calcium in larger mesenteric arteries, which require relatively high pressures for a myogenic response. This would be a significant result on its own if there were more data. Furthermore, what is swept under the rug, is that the pressure-induced constrictions of cerebral arteries are only attenuated in the Gα_12_/_13_ KO, and there are no accompanying calcium or membrane potential measurements. Possible explanations should be discussed, e.g. more robust engagement of pressure-induced depolarization/calcium elevation in cerebral arteries. This should be discussed. It is possible that a balance between two mechanisms may determine the sensitivity to pressure.

We agree with the reviewer that this should be made clearer in the Discussion. We also mention that, in contrast to mesenteric arteries, cerebral arteries do not show complete loss of myogenic vasoconstriction in the absence of smooth muscle G_12_/G_13_. This indeed may be due to a more central involvement of pressure induced depolarization/calcium elevation in cerebral arteries compared to mesenteric arteries. This is now also discussed in the third paragraph of the Discussion section.

3) Image of brain perfusion in SM-q/11-KO mice does not appear to be representative of summary data (overall impression of high intensity areas is closer to that of the other KOs than it is to the control). Should replace image with one that is more representative and provide additional detail on how images were quantified.

Thank you for making us aware of that. We are now showing a more representative image. We have also added more details on how images were quantified in the Materials and methods section (see subsection “Laser speckle perfusion measurements”).

4) G_12/13_ would presumably need to be activated in order to signal to RhoA and downstream response to inducers of myogenic constriction. It is not clear how this pathway is turned on. Ex vivo the authors use increasing vascular pressure. While it is possible that there is mechanical input the authors state the more likely physiological mechanism in their conclusion, indicating that "RhoA/Rho kinase mediated signaling is induced through G_12/13_ coupled receptors". The schematic also shows black dots which are unlabelled but perhaps meant to connote some GPCR ligand? The authors should clarify what they think is happening ex vivo, as well as in vivo where absence of G_12/13_ (thus presumably loss of signaling through a GPCR coupled to it) leads to changes in perfusion and ultimately alterations in cardiac function. The literature, or the authors own findings, using KOs for G_12/13_ linked receptors such as S1P, TXA2, PAR1 should be reviewed and used to provide a more scholarly discussion of myogenic vascular control mechanisms.

We have not performed experiments to identify the G-protein-coupled receptors involved in myogenic tone regulation in mesenteric and cerebral arteries. Several studies have shown particular GPCRs to be involved in these responses, and we are now discussing this aspect in an own paragraph in the Discussion (see Discussion, second paragraph). We also have added some more explanation to the schematic in order to better describe the principal mechanisms of GPCR involvement in myogenic vasoconstriction.

5) The schematic in Figure 5 leaves out some of the molecules involved in the transduction pathways under study. In light of the particular focus on the RhoA/Rho Kinase pathway, the myosin binding subunit MYPT-1 which is not depicted is the known substrate of Rho kinase and it has been widely used as a readout for activation of this pathway. Phospor-Abs are available. Evidence that this substrate is regulated – both increased in response to transmural pressure and lost in the G_12/13_ knockout, should be provided. Measurement of myosin phosphorylation is also possible but less useful since it reflects both the Ca increase in MLCK and Rho-mediated decrease in MLC phosphatase. The schematic in Figure 5 could be enhanced.

We have improved the schematic in Figure 6 (we assume this is the figure the reviewer meant) to better show the mechanism of Rho-kinase-regulated myosin phosphatase activity. We have also performed additional experiments to determine MYPT-1 phosphorylation in pressurized arteries from wild-type and Sm-12/13-KO mice. We found the MYPT-1 phosphorylation seen in response to induction of myogenic vasoconstriction of mesenteric arteries from wild-type mice was not seen in vessels from Sm-12/13-KO animals. These data are now presented in Figure 1G.

6) The Myh7, Acta1, and other fibrosis-related genes were not increased in the hypertrophied heart of the Sm-12/13 mice. How do cardiomyocytes become hypertrophic without an increase of contractile proteins? The authors should discuss this point.

What we measured was actually the relative expression of several genes, including Myh7 and Acta1 compared to Gapdh. Thus, the total amount of contractile proteins may well increase. An increase in the relative expression of skeletal muscle actin (ACTA1) and of the -myosin heavy chain (MYH7), which is predominant in the embryonic heart, is a hallmark of a pathological hypertrophy, whereas a physiological hypertrophy goes along with reduced relative expression of these genes or no expression change (Song et al., PLoS ONE 7, e35552 (2012); Nakamura and Sadoshima, 2018). We now discuss this point in the Discussion (see Discussion, fourth paragraph), and conclude that the cardiac hypertrophy seen in our model resembles physiological hypertrophy.

7) What are the consequences of hyper-perfusion of blood in kidney and brain? Is urine volume comparable between the Sm-12/13 and control mice? Did the authors check the sleep-wakefulness? Although the functional PET imaging is unnecessary, those regular physiological examinations might be done by the authors?

We have performed some basal analysis of kidney and brain function in Sm-12/13-KO and wild-type mice. This includes determination of urine volume, glomerular filtration rate, blood urea nitrogen and plasma creatinine levels. We also analyzed circadian activity to determine sleep-wakefulness as well as total activity and performed rotarod, bar cross and mesh grip tests to evaluate motoric functions. In none of the tests did we observe a significant difference between both groups. Thus, we conclude that the organ overperfusion within the time period studied does not lead to major kidney or brain dysfunction. These data are now presented in Figure 2—figure supplement 1.

8) The main target of RhoA is likely to be Rho-kinase in the regulation of vascular resistance in the present study; however, there has no obvious data in the present study. Therefore, the authors should tone down the description in the Discussion section.

We have in fact used the Rho-kinase inhibitor Y-27632 and tested its effect on myogenic vasoconstriction in mesenteric arteries (see Figure 1F). Y-27632 completely blocked myogenic vasoconstriction. This has already been shown in other vessels before (e.g. Gokina et al., 1985; Dubroca et al., 2005). Thus, we conclude that Rhokinase plays a critical role downstream of G_12_/G_13_ and RhoA. However, this does not exclude other signaling pathways being involved downstream of G_12_/G_13_.

9) Can the authors overexpress constitutive active RhoA in the vascular smooth muscle in the Sm-12/13 KO to see whether high output is changed or unchanged by the vascular tones? (this is not a mandatory experiment).

We have not performed this experiment. We are also skeptical whether overexpression of a constitutively active small GTPase would be an appropriate tool since the overactivation of RhoA signaling will certainly lead to various other effects, which would indirectly affect vascular tone. One might think of using an inducible system, which allows to titrate the levels of constitutively active RhoA. However, this is a technically very challenging approach, and it would probably require about a year to establish it.